# Microglia-Based Sex-Biased Neuropathology in Early-Stage Alzheimer’s Disease Model Mice and the Potential Pharmacologic Efficacy of Dioscin

**DOI:** 10.3390/cells10113261

**Published:** 2021-11-22

**Authors:** Xiao Liu, Qian Zhou, Jia-He Zhang, Ke-Yong Wang, Takashi Saito, Takaomi C. Saido, Xiaoying Wang, Xiumei Gao, Kagaku Azuma

**Affiliations:** 1Department of Anatomy, School of Medicine, University of Occupational and Environmental Health, 1-1 Iseigaoka, Yahatanishi-ku, Kitakyushu 807-8555, Japan; liuxiaoxiao0103@163.com (X.L.); zhou@med.uoeh-u.ac.jp (Q.Z.); jiahe@med.uoeh-u.ac.jp (J.-H.Z.); 2State Key Laboratory of Modern Chinese Medicine, Tianjin University of Traditional Chinese Medicine, Tianjin 301617, China; 3College of Traditional Chinese Medicine, Tianjin University of Traditional Chinese Medicine, Tianjin 301617, China; 4Shared-Use Research Center, School of Medicine, University of Occupational and Environmental Health, 1-1 Iseigaoka, Yahatanishi-ku, Kitakyushu 807-8555, Japan; kywang@med.uoeh-u.ac.jp; 5Laboratory for Proteolytic Neuroscience, RIKEN Center for Brain Science, Saitama 351-0198, Japan; saido@brain.riken.jp (T.S.); saito-t@med.nagoya-cu.ac.jp (T.C.S.); 6Department of Neurocognitive Science, Institute of Brain Science, Nagoya City University Graduate School of Medical Sciences, Nagoya 467-8601, Japan

**Keywords:** Alzheimer’s disease, *A*
*pp^NL-G-F^*, dioscin, amyloid-β, synaptic dysfunction, microglia, neuroinflammation

## Abstract

Alzheimer’s disease (AD), the most common form of dementia, is characterized by amyloid-β (Aβ) accumulation, microglia-associated neuroinflammation, and synaptic loss. The detailed neuropathologic characteristics in early-stage AD, however, are largely unclear. We evaluated the pathologic brain alterations in young adult App knock-in model *App^NL-G-F^* mice at 3 and 6 months of age, which corresponds to early-stage AD. At 3 months of age, microglia expression in the cortex and hippocampus was significantly decreased. By the age of 6 months, the number and function of the microglia increased, accompanied by progressive amyloid-β deposition, synaptic dysfunction, neuroinflammation, and dysregulation of β-catenin and NF-κB signaling pathways. The neuropathologic changes were more severe in female mice than in male mice. Oral administration of dioscin, a natural product, ameliorated the neuropathologic alterations in young *App^NL-G-F^* mice. Our findings revealed microglia-based sex-differential neuropathologic changes in a mouse model of early-stage AD and therapeutic efficacy of dioscin on the brain lesions. Dioscin may represent a potential treatment for AD.

## 1. Introduction

According to the 2018 World Alzheimer Report, a diagnosis of dementia is made every 3 s worldwide, and approximately 152 million people will be living with dementia by 2050. Alzheimer’s disease (AD) is the most common form of dementia, currently affecting more than 50 million people in the world. The disease robs people of their independence and is the fifth leading cause of death [1]. Although separate diagnostic recommendations exist for preclinical, mild impaired cognition, and dementia stages of AD, they are intended for observational and intervention studies, and not clinical care [2]. Recent evidence suggests that the appearance and deposition of amyloid-β (Aβ) is inextricably associated with neuroinflammation in the early stages of the disease, and this phenomenon manifests differently according to the stage [3]. Neuroinflammation is characterized by microglia recruitment and activity in early-stage AD, followed by synaptic dysfunction and further Aβ deposition to form plaques [4,5].

A lot of evidence indicates that Aβ accumulation and aggregation are key initiating events in the pathogenesis of AD [6]. An imbalance between production and clearance of Aβ is the main cause of Aβ aggregation in the brain, which provokes many subsequent pathologic events [7]. Increased Aβ leads to neuroinflammation and neuronal loss, and impairs synaptic activity, which in turn triggers neurodegenerative lesions [8,9,10]. Moreover, Aβ triggers a pathophysiologic cascade that leads to intracellular tau-containing neurofibrillary tangles (NFTs), both of which interact and may independently accelerate the development of brain lesions [11]. The first-generation AD mouse model was established on the basis of the overexpression of amyloid precursor protein (App) [12,13], but non-physiologic overexpression of App also gives rise to other phenotypes that are not associated with AD [14]. To address this problem, Saito et al. established second-generation App knock-in (App-KI) mice, transgenic mice carrying the Swedish (KM670/671NL), Arctic (E693G), and Bay/Iberian (I716F) mutations [15]. Plaque deposition in homozygous *App^NL-G-F^* mice begins at 3 months of age and behavioral impairment appears after 6 months [15,16,17,18,19]. The AD pathology of mushroom spine reduction and neuroinflammation are also observed in *App^NL-G-F^* mice, indicating that the pathophysiologic changes are induced by Aβ toxicity [20,21]. Thereby, this model avoids confusion due to non-physiologic signals in therapeutic test trials and can be used regardless of whether the cause of functional changes is altered by NFTs.

Preclinical AD patients with normal cognitive function have a broader time-frame for prevention and treatment, which are more effective before significant cognitive and behavioral abnormalities develop. Natural products isolated from herbal medicines have a broad range of pharmacologic properties and play a crucial role in preventing and treating various kinds of diseases. Dioscin is widely found in Leguminosae, Dioscoreaceae, Liliaceae, and Solanaceae plants [22]. As an important natural steroidal saponin component, it not only has good anti-tumor, anti-inflammatory, hypoglycemic, and hypolipidemic effects [23,24,25], it also has certain neuroimmune effects and can prevent cognitive impairment in AD mice to some extent [26]. Dioscin ameliorates neuronal damage caused by neuroinflammation and oxidative stress [26] and attenuates Aβ deposition and NFT accumulation in the cortex and hippocampus of 5 × FAD mice [27], highlighting its potential anti-AD pharmacologic efficacy.

Here, we focused on early-stage AD at 2 time-points before the emergence of behavioral deficits in young adult *App^NL-G-F^* mice (3 and 6 months of age) to explore the pharmacologic efficacy of dioscin. Brain microglia are considered key controllers of adult brain function via their modulation of microglia-neuron interactions [27]. Therefore, we first explored the pathologic brain alterations in young adult *App^NL-G-F^* mice at the 2 time-points as a model of early-stage AD. We speculate that microglial dysfunction could induce the development of AD through impairments in microglia-neuron cross-talk. Our findings suggested that pathologic brain alterations were exacerbated as age increased, with female mice exhibiting more severe alterations than males. We further investigated the effects of dioscin as a potential therapeutic agent to ameliorate the pathologic alterations.

## 2. Materials and Methods

### 2.1. Animals and Drug Administration

*App^NL-G-F^* mice were established as described previously [15], and 3- and 6-month-old male and female mice were used for the experiments. The transgenic experiment was evaluated and approved by the local committee (DP190007, 2 October 2019). Wild-type (WT) C57BL/6 mice were used for negative littermates. Mice were housed under the Laboratory Animal Research Center of University of Occupational and Environmental Health, Japan. Dioscin (Dio, Chengdu Must Bio-Technology Co., Ltd., Chengdu, China) was dissolved in olive oil by heating at 60 °C and stirring with a glass rod. Memantine hydrochloride (Mema, Tokyo Chemical Industry Co., Ltd., Portland, OR, USA), the FDA-approved drug for treating AD, was dissolved in ultrapure water. At 1 month of age, WT and *App^NL-G-F^* mice were administered olive oil (10 mL/kg), Dio (0.2 μmol/kg/day of weight), or Mema (200 μmol/kg/day of weight) by oral gavage for 2 and 5 months, respectively. All animal protocols were approved by the Ethics Committee for Animal Care and Use of the University of Occupational and Environmental Health, Japan (AE19-018, permission code, 11 November 2019).

### 2.2. Tissue Preparation

Twenty-four hours after the last drug administration, the brains of the mice were rapidly removed, washed in 0.9% cold aqueous NaCl (Otsuka Pharmaceutical, Tokyo, Japan), and dried slightly before measuring the wet weight. The cortex and hippocampus were dissected out, and the remaining brain tissue was placed in 10% formalin neutral buffer solution (FUJIFILM Wako Pure Chemical Corporation, Osaka, Japan) for histopathology examination. The cortex and hippocampus were rapidly frozen in liquid nitrogen and transferred to a −80 °C refrigerator 24 h later for subsequent experiments.

### 2.3. Golgi Staining and Dendritic Spine Analysis

Golgi staining was carried out using the superGolgi Kit (Bioenno Tech, LLC, Santa Ana, CA, USA). Briefly, the mice were anesthetized and the intact brains were removed. The brains were immersed in the impregnation solution. The solution was replaced the next day and the brains were immersed 10 or 12 additional days in the dark, and then transferred to the post-impregnation buffer for 48 h in the dark. Coronal sections (80 μm thick) were cut using a microtome (HM 355S, Thermo Fisher Scientific, Houston, TX, USA). Sections were mounted on adhesive microscope slides using staining solution and post-staining buffer, cleaned, and cover slipped using Permount (FUJIFILM Wako Pure Chemical Corporation). Dendrites of cortical neurons were examined using the Eclipse Ni-U microscope (Nikon Corporation, Tokyo, Japan), and the images were captured with a CCD camera (DS-Fi2, Nikon Corporation, Tokyo, Japan).

For quantification of spines, all images were converted to 8-bit, inverted, and background subtracted using Fiji ImageJ (National Institutes of Health, Bethesda, MD, USA) software, and dendritic spine density was measured by two independent researchers. In each animal, three intact dendritic spines of cortical neurons were selected for more detailed observation.

### 2.4. Histologic Observations

The brains were fixed in 4% paraformaldehyde or 10% neutral buffered formalin, followed by gradient ethanol dehydration. Paraffin sections were stained with hematoxylin-eosin (HE) or Congo Red (ScyTek Laboratories, Inc., Logan, UT, USA). All sections were observed under an Eclipse Ni-U microscope and a Nikon DS-Fi2 camera was used to capture the images.

### 2.5. Transmission Electron Microscopic Observations

Fresh hippocampal CA1 area tissues were obtained under a microscope and fixed with 2.5% glutaraldehyde/0.1 MPB (pH 7.4) (Nisshin EM Co., Ltd., Tokyo, Japan) at 4 °C overnight. The tissue pellets were post-fixed with 1% osmium oxide (Kishida Chemical Co., Ltd., Osaka, Japan) for 1 h. Next, the samples were dehydrated in a graded series of acetone (FUJIFILM Wako Pure Chemical Corporation). Specimens were embedded in epoxy Epon 812 (Nisshin EM Co., Ltd., Tokyo, Japan). The ultrathin sections (80 nm thick) were obtained with a diamond knife using an ultramicrotome (Leica EM UC7, Wetzlar, Germany), stained with uranyl acetate and lead salts, and examined under a transmission electron microscope (JEM-1400Plus, JEOL Ltd., Tokyo, Japan).

### 2.6. Immunohistochemistry

Paraffin sections (4 μm) were routinely dewaxed and hydrated with xylene and ethanol. Antigen retrieval (Dako, Glostrup, Denmark) was performed by autoclaving for 10 min. Endogenous peroxidase was inactivated by 3% H_2_O_2_ (FUJIFILM Wako Pure Chemical Corporation) and blocking solution (Dako). Sections were incubated with diluted primary antibody for 1 h. Subsequently, sections were incubated with secondary antibody for 30 min (Nichirei Corporation, Tokyo, Japan). After staining with diaminobenzidine (FUJIFILM Wako Pure Chemical Corporation) and counterstaining with hematoxylin, the images were recorded using an Eclipse Ni-U microscope and captured using a Nikon DS-Fi2. The primary antibodies used are summarized in Table 1.

### 2.7. Immunofluorescence

Paraffin sections (4 μm) were routinely dewaxed and hydrated with xylene and ethanol, followed by antigen retrieval. The sections were incubated for 10 min in blocking solution and then with diluted primary antibody for 1 h. The primary antibodies were visualized with Alexa Fluor 488 F(ab’)2-goat anti-Rabbit IgG (H+L), Alexa Fluor 546 donkey anti-goat IgG (H+L), and Alexa Fluor 546 goat anti-mouse IgG (H+L) (1:200; Thermo Fisher Scientific, Waltham, MA, USA) for 1 h. Images were captured with an Olympus VS120 camera (Olympus, Tokyo, Japan) and digitized with OlyVIA software (Olympus). Single images were converted to 12-bit TIFF images.

### 2.8. Western Blot Analyses

Cerebral cortex and hippocampal tissues were lysed with RIPA buffer (Millipore Corp, Bedford, MA, USA) and centrifuged at 12,000× *g* for 30 min at 4 °C, and the supernatant was collected. The protein concentration was evaluated with the BCA protein assay kit (Thermo Fisher Scientific). The protein was isolated in 4–12% Bis-Tris Gel (Thermo Fisher Scientific) and transferred to a polyvinylidene difluoride membrane (Millipore). After blocking with 5% skim milk or 3% bovine serum albumin for 2 h, TBST was used to dilute the first antibody and the mixture was incubated overnight at 4 °C. After washing with TBST, the tissues were then incubated with the secondary antibodies (Cell Signaling Technology, Danvers, MA, USA) for 1 h. An ECL kit (Cytiva, Buckinghamshire, UK) was used to detect the chemiluminescence signal of protein, and ImageJ software was used for quantification.

### 2.9. ELISA Assay

Blood samples were collected from the medial canthus of the mice and centrifuged at 3000 rpm for 10 min at 4 °C to separate the serum. The serum IL-1β and TNF-α levels were measured using an enzyme-linked immunosorbent assay (ELISA) kits, according to the manufacturer’s instructions (Elascience Biotechnology, Huston, TX, USA). Light absorbance was detected by an absorbance microplate reader (Corona Electric Co. Ltd., Ibaraki, Japan).

### 2.10. Statistical Analyses

Statistical analyses were performed with Statistical Package for Social Science (SPSS, Chicago, IL, USA) software (Version 23.0). All data are expressed as mean ± SEM. ImageJ was used to quantify the immunohistochemical staining intensity. One-way analysis of variance was used to compare differences among the groups. The least significant difference test was used for parametric data, and Dunnett’s T3 test was applied for nonparametric data. p value of less than 0.05 was considered to be statistically significant.

## 3. Results

### 3.1. Pathologic Changes in App^NL-G-F^ Mice and Dio Treatment

Continual mouse body weight measurements obtained for 5 months revealed a gradual increase in body weight of *Ap^pNL-G-F^* mice with age compared to that of WT mice. The increase in the body weight with age was significant in females during days 20–60 of observation. The body weight of mice treated with Dio or Mema was lower than that of the intact *App^NL-G-F^* mice (Figure 1A). In addition, the brain index of *App^NL-G-F^* mice was higher than that of WT mice, and no significant changes were found in the treated groups (Figure 1B).

Amyloid deposition was determined by HE-staining and Congo red staining. In 3-month-old WT mice, there was no obvious neuronal damage; neuronal cell contours were intact, and cells were tightly arranged. In the hippocampal CA1 region of *App^NL-G-F^* mice, the cytoplasm appeared dark with small, condensed nuclei. No significant differences were found between male and female *App^NL-G-F^* mice, and reversal of the changes was observed after administration of Dio and Mema (Figure 1C,D). *App^NL-G-F^* male mice at 6 months showed a significant number of degenerative neurons in the dentate gyrus and CA1 region under low magnification. The number of degenerative neurons was significantly reduced, however, by the administration Dio and Mema. There was no significant change in *App^NL-G-F^* female mice at 6 months regardless of drug administration (Figure 1E,F).

To further determine whether amyloid deposition occurs in *App^NL-G-F^* mice at an early stage, Congo red staining was evaluated. The 3-month-old mice showed no pathologic changes. Brick-red plaques appeared in both the cortex and hippocampus of 6-month-old *App^NL-G-F^* mice, with specific apple-green protein deposits observed under polarized light (Figure 1G,H), indicating the beginning of amyloid deposition in terms of pathology and relatively small cell volumes with low densities. Treatment with Dio and Mema apparently improved the pathologic alterations.

### 3.2. Dio Inhibits the Abnormal Accumulation of Aβ Precursors and Alleviates Aβ Burden in Early-Stage AD Model Mice

We examined the two main factors of App and β-amyloid precursor protein cleaving enzyme (BACE) associated with Aβ production. We found no significant difference in App expression of the cerebral cortex and hippocampus of WT and *App^NL-G-F^* mice, except in the hippocampus of 3-month-old female mice. The expression of BACE and Aβ, however, was significantly increased in the cerebral cortex and hippocampus of *App^NL-G-F^* mice at both 3 and 6 months of age compared with WT mice. The expression of BACE and Aβ was significantly decreased in *App^NL-G-F^* mice after Dio treatment, and this effect was superior to that of Mema (Figure 2A–D).

To examine the Aβ burden in *App^NL-G-F^* mice, the brain sections were immunostained and the amount of Aβ plaque was quantified. Compared with WT mice, Aβ immunoreactivity was visible in the male and female *App^NL-G-F^* mice in cortex and hippocampus at 3 months of age. The Aβ plaque area increased by 6 months of age (Figure 2E,F). Interestingly, a higher percentage of plaque area was observed in female mice than in male mice from 3 months of age onwards in both the cortex and hippocampus (Figure 2G). The mice treated with Dio and Mema showed a remarkable decrease in Aβ deposition in the cortex and hippocampus.

### 3.3. Dio Alters Synaptic Dysfunction in Young Adult App^NL-G-F^ Mice

In *App^NL-G-F^* mice, the age before 6 months is considered to be a pre-symptomatic stage as it precedes the signs of deficits in spatial memory and learning [28]. We initially observed the ultrastructure of hippocampal synapses by electron microscopy. We observed no significant changes in the synaptic microstructure in 3-month-old *App^NL-G-F^* and WT mice. Compared with WT mice, the postsynaptic densities were shorter and thinner, and the presynaptic vesicles were decreased in 6-month-old *App^NL-G-F^* mice (Figure 3F,K). These findings suggested that by 6 months of age, the inter-synaptic transmission function of *App^NL-G-F^* mice is impaired. Next, we evaluated the dendritic spines in the cortex of 3- and 6-month-old *App^NL-G-F^* mice. According to the Golgi staining results, at 3 months of age, the dendritic spines of male *App^NL-G-F^* mice were not damaged. In female *App^NL-G-F^* mice, however, the number of dendritic spines, especially mushroom-shaped spines, was significantly decreased. By 6 months of age, *App^NL-G-F^* mice of both sexes showed a significant decrease in dendritic spines. Dio and Mema significantly increased the total spine density in females (Figure 3A–C).

The immunoreactivity of postsynaptic density protein 95 (PSD95) and synaptophysin (SYP) in the cortex and hippocampus was measured by immunofluorescence staining. The fluorescence intensity of PSD95 and SYP was weaker in the *App^NL-G-F^* mice of both sexes compared with WT mice, and more pronounced at 6 months of age. Dio or Mema treatment increased PSD95 and SYP immunoreactivity (Figure 3D,E,I,J). These results were consistent with the findings of the Golgi staining. To further evaluate synaptic plasticity in *App^NL-G-F^* mice, we measured the protein expression of PSD95 and SYP by Western blotting. The results showed that PSD95 expression was reduced in the cortex and hippocampus of 3-month-old male and female *App^NL-G-F^* mice and increased after administration with Dio and Mema, while no significant alterations were found in the hippocampus of male mice. No changes in the expression of SYP were observed (Figure 3G,H). In 6-month-old *App^NL-G-F^* mice, the expression of both PSD95 and SYP was significantly decreased in the cortex and hippocampus, indicating that the synapses of *App^NL-G-F^* mice were significantly impaired, which may be associated with the beginning of behavioral and memory impairments. Expression of the 2 synapse-associated proteins was significantly upregulated after administration of Dio and Mema (Figure 3L,M).

### 3.4. Dio Affects Microglial Activation in Young Adult App^nl-G-F^ Mice

Microglial activation is closely related to the development and progression of AD. To examine the brain microglial alterations in *App^NL-G-F^* mice, immunostaining of brain sections was performed using the ionizing calcium binding adapter molecule (Iba1, microglia marker) antibody. We found that microglial immunoreactivity was reduced in the cortex and hippocampus of 3-month-old male and female *App^NL-G-F^* compared with WT mice. By 6 months of age, however, microglial activity was higher in the cortex and hippocampus of *App^NL-G-F^* mice than in WT mice (Figure 4A–C). Therefore, we next investigated the ultrastructure of microglia in 6-month-old *App^NL-G-F^* mice, which showed a significant increase in lysosomes in the cytoplasm, depressed nuclear membranes, and more active microglia around the myelin sheath. Peripheral vacuole-like structures and dark granular bodies were observed (Figure 4D). We next investigated the changes in the microglia at the molecular level. We observed a trend toward significantly lower or reduced microglia expression in the cortex and hippocampus of both male and female *App^NL-G-F^* mice at 3 months of age, but no therapeutic effects of Dio or Mema. At 6 months of age, however, microglia appeared to be highly expressed. Both Dio and Mema appeared to inhibit the microglial activation (Figure 4E–G).

To determine the changes in microglia recruitment at two early time-points, we performed immunofluorescence staining. The results indicated that there were few activated microglia in the cortex and hippocampus of 3-month-old male and female *App^NL-G-F^* mice, a decrease in fluorescently labeled Aβ plaques after administration with Dio and Mema, and the appearance of labeled microglia around the plaques, indicating phagocytic activity of the microglia (Figure 4H,I). By 6 months of age, microglia recruitment around Aβ plaques was increased in the cortex and hippocampus of male and female *App^NL-G-F^* mice, and the area of fluorescently labeled Aβ plaques was decreased after administration with Dio and Mema, as was the number of activated microglia (Figure 4J,K). This finding suggests that either Dio or Mema enhances the phagocytic activity of microglia in the mice at 3 months of age, thereby reducing Aβ plaques. By 6 months of age, however, when microglia may act to promote inflammation, Dio and Mema exhibited anti-inflammatory effects.

### 3.5. Dio Regulates Neuroinflammatory Responses in Young Adult App^nl-G-F^ Mouse Brain by Affecting NF-Κb/Β-Catenin

To verity the inflammatory response, we examined the changes of pro-inflammatory cytokine levels in *App^NL-G-F^* mouse serum. For male *App^NL-G-F^* mice, the serum IL-1β level was significantly increased at 6 months of age, which tended to decrease after administration with Dio. In female *App^NL-G-F^* mice, the serum TNF-α level was significantly increased at 3 months of age. At 6 months of age, both IL-1β and TNF-α levels were significantly increased. Treatment with Dio and Mema inhibited the elevation of the serum IL-1β and TNF-α levels (Figure 5A).

When microglia are activated in large numbers, an inflammatory response emerges. Immunohistochemical staining for NF-κB and β-catenin in the cortex and hippocampus of 6-month-old male and female mice was examined and representative micrographs are shown. The fluorescence intensity of NF-κB in the cortex and hippocampus of 6-month-old *App^NL-G-F^* male and female mice was higher than that in WT mice. The β-catenin co-localized with NF-κB in the brains of both WT and *App^NL-G-F^* mice (Figure 5B,E). No significant changes in fluorescence intensity were observed after administration with Dio and Mema. These co-localizations suggested the possibility of some mutual regulation between NF-κB and β-catenin signaling pathways. β-catenin protein expression was reduced in the cortex and hippocampus of 6-month-old *App^NL-G-F^* mice of both sexes, and the expression increased after administration with Dio and Mema. In contrast, the ratio between phosphorylated glycogen synthase kinase 3β (GSK-3β, Ser9) and GSK-3β, and phosphorylated NF-κB p65 and NF-κB p65 was higher in *App^NL-G-F^* mice, and the protein expression ratio was decreased after the administration with either Dio or Mema (Figure 5C,D,F,G).

## 4. Discussion

Synaptic dysfunction and neuroinflammation are important events in the pathophysiology of AD. The pathologic changes preceding the development of behavioral disorders in *App^NL-G-F^* mice of both sexes, however, are not completely elucidated. Our findings showed that Aβ deposition gradually increased with age from 3 months in both male and female *App^NL-G-F^* mice. The area of plaque deposition, accompanied by synaptic damage, was significantly larger in females than in males. In addition, the PSD95 protein expression levels were significantly decreased in the cerebral cortex and hippocampus at 3 months of age. The Aβ plaque deposition in the brains of *App^NL-G-F^* mice was associated with neuroinflammation. Interestingly, here, the role of microglia may have switched. The pathologic aggravation of AD also leads to abnormalities in the β-catenin signaling pathway, and Dio is effective as a preventive drug. This is the first report of the early neuropathologic changes of AD model App mice of both sexes. Furthermore, we demonstrated that Dio is a natural compound with potential therapeutic effects targeting Aβ accumulation.

Mema is a non-competitive N-methyl-D-aspartate receptor antagonist used for treating AD. Animal studies exhibited that Mema could counteract neuropathologic alterations in a preclinical mouse of AD [29]. In the present study we selected Mema for comparison. We found for the first time that Mema had inhibitory effects on Aβ accumulation.

The body weights of *App^NL-G-F^* mice were recorded from 1 month to 6 months of age, and were higher than those of WT mice, especially in female mice. By 6 months of age, the body weights were not significantly different between *App^NL-G-F^* and WT mice. A recent study showed that female *App^NL-G-F^* mice at 3 months of age had decreased motility of the small intestine compared with WT mice [30]. The decreased motility may lead to food remaining longer in the intestine, thus altering digestion and absorption. Therefore, the weight gain was much greater in 3-month-old female *App**^NL-G-F^* mice. Other factors, however, cannot be excluded, including control of food and analysis of metabolite collection, as well as the emergence of behavioral disorders or disease progression triggering other pathologic mechanisms that may contribute more to body weight gain.

Among the factors associated with Aβ production, BACE was significantly highly expressed, while APP did not change significantly in *App^NL-G-F^* male and female mice. Aβ deposition was already present in *APP^NL-G-F^* male and female mice at 3 months old, and both immunoreactivity and protein expression were significantly higher in *App^NL-G-F^* than in WT mice of the same sex. Notably, the intensity of the region of Aβ deposition in the same age group remained greater in female mice than in male mice. This indicates that sex differences diverged from the very beginning of Aβ deposition [16]. Aβ has synaptotoxic effects and can induce synaptic loss [31]. Memory loss in AD is caused by “synaptic failure” [21,32,33]. In the early stage of disease progression, when behavioral changes in AD model mice are relatively mild, synaptic loss is already beginning [15,34], which is consistent with our observations. The absence of mushroom spines may underlie cognitive deficits during AD progression [21,34]; SYP and PSD95 are commonly used to label pre- and post-excitatory synaptic structures. We found sex differences already at early time-points, that is, significant loss of mushroom spines in both the cerebral cortex and hippocampus of 3- and 6-month-old female *App^NL-G-F^* mice, and significantly decreased expression of PSD95, suggesting that female *App^NL-G-F^* mice experience earlier memory and cognitive decline. Dio treatment had a significant restorative effect on PSD95 and SYP expression in the cerebral cortex and hippocampus of *App^NL-G-F^* mice, suggesting that Dio may further restore synaptic impairment by reducing Aβ plaques and decreasing their toxicity.

The observed changes in the microglia of *App^NL-G-F^* mice were interesting. Low expression of microglia was observed in the cerebral cortex and hippocampus of *App^NL-G-F^* mice at 3 months of age, whereas the expression was higher in *App^NL-G-F^* mice than in WT mice of the same sex at 6 months of age, a finding that is inconsistent with changes observed in other AD model mice [35]. Microglia have diverse critical functions, including monitoring neuronal activity, modulating cognition, and phagocytosis. Whether microglia-associated neuroinflammation is beneficial or detrimental during the progression of AD depends on the microenvironment. Our findings revealed an inconsistent microglia status at two different early time-points. On this basis, we hypothesize that the central microglia are very scarce in *App^NL-G-F^* mice, which may trigger the initiation of various neuropathologic alterations from the earliest stage of Aβ accumulation. Aβ deposition from 3 to 6 months of age is concomitant with the activation of microglial phagocytosis around Aβ plaques to increase Aβ clearance and prevent the development of AD pathology. Observations during this phase show that microglia are in a form that is antagonistic to disease progression. It is worth noting, however, that persistent aggregation of microglia leads to the release of cytotoxic molecules such as pro-inflammatory factors, reactive oxygen species, chemokines, and complement components [36,37,38], which can cause microglial dysfunction and an “overhealing” phenomenon. After Dio administration, immunostaining showed an increase in the number of Iba1-positive cells in 3-month-old *App^NL-G-F^* male and female mice, without significant changes in the protein expression levels, which was not consistent with a role of Dio to promote microglial activation for phagocytosis. At 6 months of age, the number of positive microglia and the recruitment of microglia around the Aβ plaques was decreased in the Dio group compared with the *App^NL-G-F^* group; protein expression was also significantly decreased, which suggests that Dio inhibits the abnormal function of microglia. Therefore, in contrast to some components of ginseng, Dio did not exhibit immune “bidirectional regulation”, but only inhibited microglia activation.

Aβ accumulation was accompanied by a neuroinflammatory response in the early stage. When *App^NL-G-F^* mice were 6 months old, microglia were recruited in large numbers, indicating an inflammatory response. Aβ induces neuroinflammation, and NF-κB in turn promotes further Aβ production [39]. Wnt signaling dysfunction also leads to Aβ production and aggregation [40,41]. Components of the classical β-catenin pathway regulate inflammatory responses and immune responses through interactions with NF-κB [42]. When Aβ begins to accumulate, GSK-3β is activated and the classical β-catenin pathway is downregulated in the pathogenesis of AD by decreasing the level of β-catenin or increasing the activity of GSK-3β and Dickkopf-1 [43,44,45]. A deeper understanding of the molecular basis of cross-regulation in chronic complex diseases will help to elucidate the underlying pathophysiologic mechanisms of related diseases and contribute to the development of more specific and effective therapeutic options [45]. Further investigation is needed to elucidate the molecular mechanisms underlying changes in the β-catenin signaling in *APP^NL-G-F^* mouse brains in response to NF-κB–mediated early inflammatory responses.

## 5. Conclusions

In summary, our findings provide evidence that young adult *App^NL-G-F^* male and female mice show progressively increased Aβ deposition, synaptic loss, neuroinflammation, and dysregulation of β-catenin signaling (Figure 6). Importantly, we uncovered a dual phenotype of the microglia response during the early stages of amyloid deposition. The pathologic processes were more severe in female mice than in males, especially in terms of synaptic loss. Therefore, sex differences in disease progression and presentation should be considered. As a natural compound, dioscin had a significant therapeutic effect on early-stage *App^NL-G-F^* mouse lesions and therefore may represent a promising drug targeting Aβ to alleviate the pathologic symptoms of early AD and delay disease progression.

## Figures and Tables

**Figure 1 cells-10-03261-f001:**
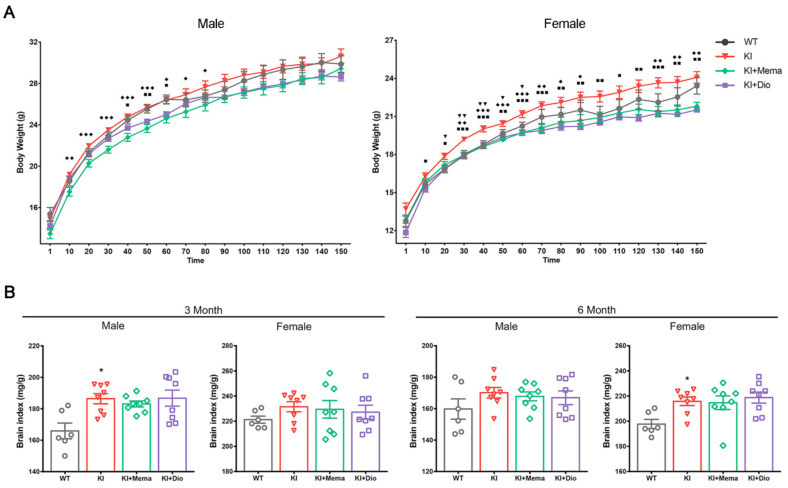
Body weight, brain index, and hippocampal morphology in young adult *App^NL-G-F^* mice. (**A**) Representative body weight changes from the first day of administration (1-month-old) to 150 days (6-month-old). Data are expressed as mean ± SEM. *n* = 12-24 mice/group. ^▼^
*p* < 0.05 compared with WT; ^▼▼^
*p* < 0.01 compared with WT. ^■^
*p* < 0.05 compared with *App^NL-G-F^*; ^■■^
*p* < 0.01 compared with *App^NL-G-F^*; ^■■■^
*p* < 0.001 compared with *App^NL-G-F^*. ^◆^
*p* < 0.05 compared with *App^NL-G-F^*; ^◆◆^
*p* < 0.01 compared with *App^NL-G-F^*; ^◆◆◆^
*p* < 0.001 compared with *App^NL-G-F^*; (**B**) Representative ratio of brain weight to body weight in male and female mice at 3 and 6 months of age. Data are expressed as mean ± SEM. *n* = 6–8 mice/group. * *p* < 0.05 compared with WT; (**C**–**F**) Representative HE-staining in male and female mice at 3 and 6 months of age. Scale bars: 1 mm (panel Hippocampus) and 0.1 mm (panels DG, CA1, and CA3). *n* = 4–5 mice/group; (**G**,**H**) Representative CongoRed staining in male and female mice at 3 and 6 months of age. Scale bar: 0.1 mm. *n* = 4–5 mice/group; (**C**) 3-month-old male mice; (**D**) 3-month-old female mice; (**E**,**G**) 6-month-old male mice; (**F**,**H**) 6-month-old female mice.

**Figure 2 cells-10-03261-f002:**
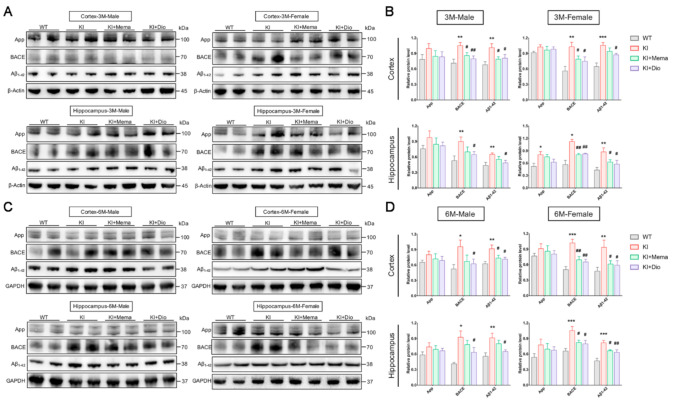
Effect of Dio on abnormal accumulation of Aβ in young adult *App^NL-G-F^* mice. (**A**–**D**) Western blotting and quantification of App, BACE, and Aβ protein expression in the cortex and hippocampus of 3- and 6-month-old male and female WT or *App^NL-G-F^* mice. Data are expressed as mean ± SEM. *n* = 6 mice/group. * *p* < 0.05 compared with WT; ** *p* < 0.01 compared with WT; *** *p* < 0.001 compared with WT. ^#^
*p* < 0.05 compared with *App^NL-G-F^*; ^##^
*p* < 0.01 compared with *App^NL-G-F^*; (**E**–**G**) Images and quantification showing the Aβ plaque staining in the cortex and hippocampal CA1 region in male and female WT and *App^NL-G-F^* mice at 3 and 6 months of age. Scale bar: 0.1 mm. Data are expressed as the mean ± SEM. *n* = 3 slides/group. Four randomly selected fields of view for each section under the microscope. *** *p* < 0.001 compared with WT. ^#^
*p* < 0.05 compared with *App^NL-G-F^*; ^##^
*p* < 0.01 compared with *App^NL-G-F^*; ^###^
*p* < 0.001 compared with *App^NL-G-F^*.

**Figure 3 cells-10-03261-f003:**
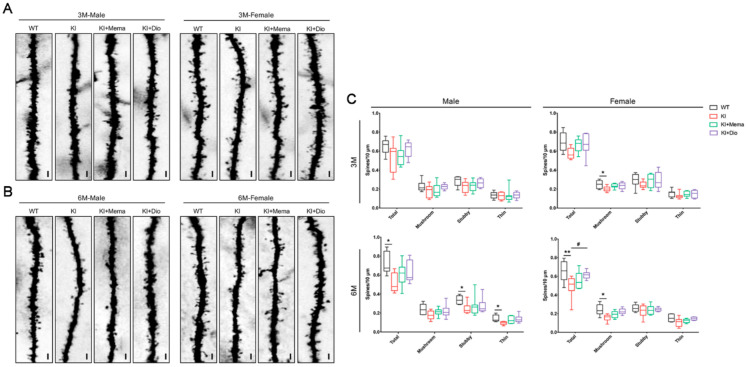
Effect of Dio on synaptic loss in young *App^NL-G-F^* mice. (**A**–**C**) Representative Golgi-staining and quantification showing spines of 3- and 6-month-old male and female WT and *App^NL-G-F^* mice. Scale bar: 0.1 mm. Data are expressed as mean ± SEM (*n* = 8). * *p <* 0.05 compared with WT; ** *p <* 0.01 compared with WT. ^#^
*p <* 0.05 compared with *App^NL-G-F^*; (**F**,**K**) TEM image from male and female WT and *App^NL-G-F^* mice showing synapses. Scale bar: 0.5 μm; (**F**) 3-month-old male and female mice; (**K**) 6-month-old male and female mice. (**D**,**E**,**I**,**J**) Immunofluorescence staining of PSD95 and SYP in the cortex and hippocampus in 3- and 6-month-old male and female WT and *App^NL-G-F^* mice. Scale bar: 0.1 mm; (**D**) 3-month-old male mice; (**E**) 3-month-old female mice; (**I**) 6-month-old male mice; (**J**) 6-month-old female mice. (**G**,**H**,**L**,**M**) Western blot and quantification of PSD95 and SYP protein expression in the cortex and hippocampus in 3- and 6-month-old male and female WT and *App^NL-G-F^* mice. Data are expressed as mean ± SEM. *n* = 6 mice/group. * *p* < 0.05 compared with WT; ** *p* < 0.01 compared with WT; *** *p* < 0.001 compared with WT. ^#^
*p* < 0.05 compared with *App^NL-G-F^*; ^##^
*p* < 0.01 compared with *App^NL-G-F^*.

**Figure 4 cells-10-03261-f004:**
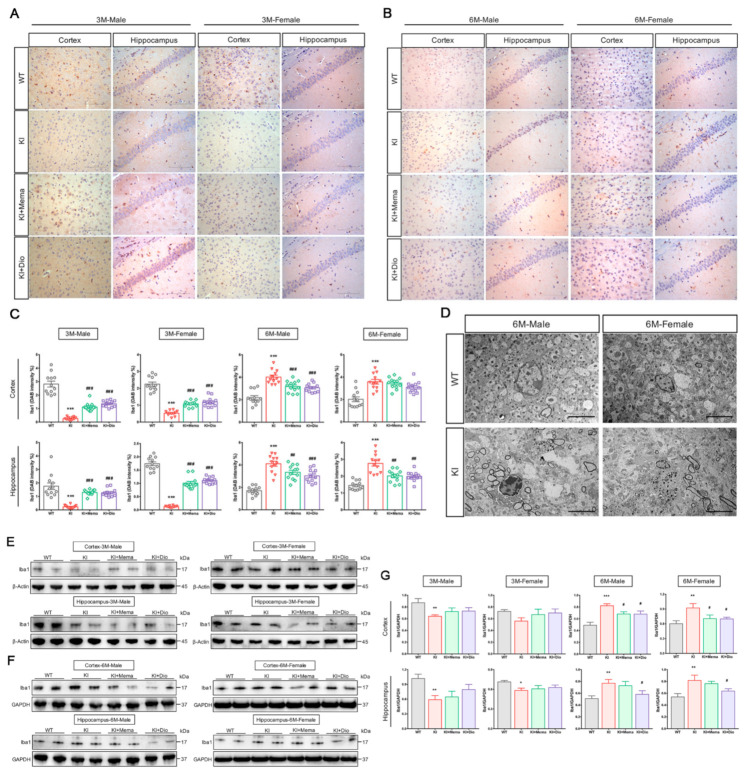
Effect of Dio on microglial activity in young *App^NL-G-F^* mice. (**A**–**C**) Images and quantification showing Iba1 staining in the cortex and hippocampal CA1 region in 3 and 6-month-old male and female WT and *App^NL-G-F^* mice. Scale bar: 0.1 mm. Data are expressed as mean ± SEM. *n* = 3 slides/group. Four randomly selected fields of view for each section under a microscope. *** *p* < 0.001 compared with WT. ^##^
*p* < 0.01 compared with *App^NL-G-F^*; ^###^
*p* < 0.001 compared with *App^NL-G-F^*; (**D**) TEM image from 6-month-old male and female WT and *App^NL-G-F^* mice showing microglia. Scale bar: 5 μm; (**E**–**G**) Western blot and quantification of Iba1 protein expression in the cortex and hippocampus in 3- and 6-month-old male and female WT and *App^NL-G-F^* mice. Data are expressed as mean ± SEM. *n* = 6 mice/group. * *p* < 0.05 compared with WT; ** *p* < 0.01 compared with WT; *** *p* < 0.001 compared with WT. ^#^
*p* < 0.05 compared with *App^NL-G-F^*; (**H**–**K**) Merged immunofluorescence images of Iba1, Aβ and DAPI staining in the cortex and hippocampus of 3- and 6-month-old male and female WT and *App^NL-G-F^* mice. Scale bars: 1 mm and 0.1 mm; (**H**) 3-month-old male mice; (**I**) 3-month-old female mice; (**J**) 6-month-old male mice; (**K**) 6-month-old female mice.

**Figure 5 cells-10-03261-f005:**
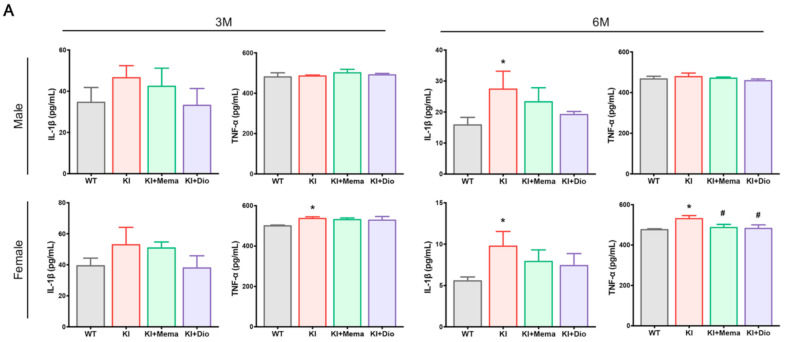
Dio affects proinflammatory cytokines and brain NF-κB/β-catenin signaling in young *App^NL-G-F^* mice. (**A**) The serum IL-1β and TNF-α levels in 3- and 6-month-old male and female WT and *App^NL-G-F^*. (**B**,**E**) Merged immunofluorescence images of NF-κB, β-catenin, and DAPI co-localized in the cortex and hippocampal region in 6-month-old male and female WT and *App^NL-G-F^* mice. Scale bar: 10 μm; (**C**,**D**,**F**,**G**) Western blot and quantification of GSK-3β, p-GSK-3β (Ser9), NF-κB p6, 5p-NF-κB p65, and β-catenin protein expression in the cortex and hippocampus in 6-month-old male and female WT and *App^NL-G-F^* mice. Data are expressed as mean ± SEM. *n* = 6 mice/group. * *p* < 0.05 compared with WT; ** *p* < 0.01 compared with WT; *** *p* < 0.001 compared with WT. ^#^
*p* < 0.05 compared with *App^NL-G-F^*.

**Figure 6 cells-10-03261-f006:**
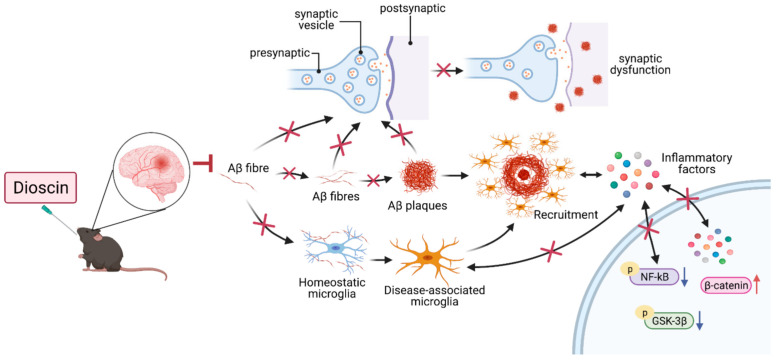
In the early stage, Aβ fibers were formed without neurological symptoms. Once formed, Aβ fibers began to accumulate and deposit around synapses and tangled around microglia to activate microglia. Activated microglia will be recruited around Aβ plate for phagocytosis, and microglia on the other side will also produce inflammatory factors to stimulate inflammatory response.

**Table 1 cells-10-03261-t001:** Antibodies used in this study.

**Antibody**	**CAT#**	**Source**	**MW (kDa)**	**Dilution**	**Application**
Anti-Aβ1-42	ab201060	Abcam	37–50	1:500/1:1000/1:200	WB/IHC/IF
Anti-Iba1	ab48004	Abcam	17	1:1000/1:250/1:200	WB/IHC/IF
Anti-PSD95	#2507	CST	95	1:1000/1:400	WB/IF
Anti-Synaptophysin	#36406	CST	38	1:1000/1:100	WB/IF
Anti-BACE	#5606	CST	70	1:1000	WB
Anti-App	#2452	CST	100–140	1:1000	WB
Anti-NF-κB p65	#8242	CST	65	1:1000	WB
Anti-pNF-κB p65	sc166748	SCB	65	1:1000	WB
Anti-GSK-3β	#9832	CST	46	1:1000	WB
Anti-pGSK-3β(Ser9)	#9323	CST	46	1:1000	WB
Anti-β-catenin	sc-7963	SCB	92	1:1000	WB
Anti-GAPDH	#2118	CST	37	1:1000	WB
Anti-β-Actin	#4970	CST	45	1:1000	WB

MW: molecular weight; CST: cell signaling technology; WB: Western blot; IF: immunofluorescence; IHC: immunohistochemistry; SCB: Santa Cruz Biotechnology.

## Data Availability

The data generated in this study is available from the corresponding author Kagaku Azuma (kazuma@med.uoeh-u.ac.jp).

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
