# Peer review of "Microglia-Based Sex-Biased Neuropathology in Early-Stage Alzheimer’s Disease Model Mice and the Potential Pharmacologic Efficacy of Dioscin"

_cells, 2021, doi:10.3390/cells10113261_

Round 1

Reviewer 1 Report

This is clearly a technically excellent study of potential great interest to the field.  It contains a LOT of data that is tightly compacted into a lot of unreadable figures.  To even hope to read them, the images had to be magnified to 250%.  Is it not possible for the authors to present their key findings in a more consise manner?

Direct issues

a)  The rationale for the choice of memantine as the "alternative" treatment for contrast with dio is not explained.  I would suggest that this is considered.

b)  I am having problems understanding the images in Figure 4.  There appear to be many more IBA positive cells in the wt mice than in the APP or APP treated mice.  I notice in some, not all, of the blots for IBA, there is increased levels in wt mice.  This seems curious and requires some discussion.

c)  Why is the molecular weight of Ab1-42 listed on blots as being 38 kD.

d)  The Ab lowering activity of memantine seems comparable to Dio.  Would the authors care to comment on this.  This seems an interesting discovery.

e)  The integration of the data from NFkb with the other measures is hard to understand.  Would the authors suggest a time course for these changes in the overall pathological mechanism.  Time course experiments would be needed.  The final figures suggests that the reduced activation is downstream from initial changes.

f)  Could the authors verify that they confirmed the specificity of the p-p65 NFkB antibody.  Any antibody from Santa Cruz has to be considered of suspect quality unless verified.  Did the authors verify with a dephosphorylation experiment.  This seems very important to confirm the validity of the proposed mechanism.

g)  The issue of describing "activated" microglia compared to resting microglia should be addressed with another marker, or at least with justification.  IBA does not describe the microglia phenotype.

Author Response

The response to reviewers’ comments

Reviewer 1

This is clearly a technically excellent study of potential great interest to the field.  It contains a LOT of data that is tightly compacted into a lot of unreadable figures.  To even hope to read them, the images had to be magnified to 250%.  Is it not possible for the authors to present their key findings in a more concise manner?

Response: We thank the reviewer for the valuable comments. All images were magnified in the manuscript, according to the reviewer’s suggestions.

Direct issues

  1. The rationale for the choice of memantine as the "alternative" treatment for contrast with dio is not explained.  I would suggest that this is considered.

Response: We thank the reviewer for raising this point. Memantine (Mema) is a non-competitive N-methyl-D-aspartate receptor antagonist used for treating Alzheimer's disease. Animal studies showed that Mema could counteract pathological alterations in a preclinical mouse model of Alzheimer's disease (Br J Pharmacol. 2012, 167, 324-52). However, its use is limited by side effects, including dizziness, headache and confusion. We examined the effects of Dioscin compared with Mema in this study. The comparison of the effects of Dioscin and Mema was added in the Results and Discussion sections, as the reviewer suggested.

  1. I am having problems understanding the images in Figure 4.  There appear to be many more IBA positive cells in the wt mice than in the APP or APP treated mice.  I notice in some, not all, of the blots for IBA, there is increased levels in wt mice.  This seems curious and requires some discussion.

Response: We appreciate this reviewer’s thoughtful comments. Ionized calcium-binding adaptor protein-1 (Iba1) is specifically and constitutively expressed in all microglia (J Histochem Cytochem 55:687-700, 2007). Microglia have diverse critical functions involved in homeostasis and host defense mechanisms. In this study we examined the Iba1-positive microglia in AppNL-G-F mice at 3 and 6 months of age, the early stage of Alzheimer's disease. We found that microglia was reduced in the cortex and hippocampus of 3-month-old AppNL-G-F compared with WT mice. By 6 months of age, however, microglia was higher in the cortex and hippocampus of AppNL-G-F mice than in WT mice (Figure 4). Our findings revealed an inconsistent microglia status at 2 different early time-points. Whether microglia-associated neuroinflammation is beneficial or detrimental during the progression of AD depends on the microenvironment. The detailed mechanism is needed to be discussed further.

  1. Why is the molecular weight of Ab1-42 listed on blots as being 38 kD.

Response: We thank the reviewer for this indication. The anti-Aβ we applied was ‘Recombinant Anti-Beta Amyloid Anti-body (ab201060)’. According to ‘ab201060’ in the latest Abcam official website, this antibody does not exist the recommendation of Western blot. The antibody was purchased in January 2020, and in the given reference blot, two bands appeared in the brain of APP/PS1 mice, including 'monomer' of 4 kDa and 'high MW oligomer' of about 38 kDa. In our study, the blot band appeared at 38 kDa. The molecular weight of Aβ in the relevant references are shown at around 36-42 kDa (PMID: 33478552, PMID: 32453744, PMID: 32595486, PMID: 30774310).

  1. The Ab lowering activity of memantine seems comparable to Dio.  Would the authors care to comment on this.  This seems an interesting discovery.

Response: We thank this reviewer for the constructive comments. Accumulation of aggregated Ab in the brain is believed to be an important pathological mechanism of Alzheimer’s disease. Our study showed that both Dio and Mema could reduce Ab accumulation in the cortex and hippocampus. The effect of Dio was superior to that of Mema (Figure 2). Therefore, we consider that Dio may represent a promising drug targeting-Ab to alleviate the pathologic symptoms of Alzheimer’s disease.

  1. The integration of the data from NFkb with the other measures is hard to understand.  Would the authors suggest a time course for these changes in the overall pathological mechanism.  Time course experiments would be needed.  The final figures suggests that the reduced activation is downstream from initial changes.

Response: We thank the reviewer for the critical comments. We examined NF-kB expression by immunohistochemistry and Western blot at 6 months of age. We found that NF-kB expression was significantly higher in AppNL-G-F mice. Administration with Dio and Mema significantly inhibited the elevation of NF-kB expression. However, we did not measure NF-kB expression at 3 months of age. We speculate the alteration of NF-kB expression at 3 months might be less noticeable than that at 6 months. It is better to have a time course for changes of NF-kB, as the reviewer indicated.

We revised the relationship between the inflammatory factors and NF-kB at the bottom left of the Figure 6, the final figure for avoiding misunderstanding, as the reviewer indicated.

  1. Could the authors verify that they confirmed the specificity of the p-p65 NFkB antibody. Any antibody from Santa Cruz has to be considered of suspect quality unless verified.  Did the authors verify with a dephosphorylation experiment.  This seems very important to confirm the validity of the proposed mechanism.

Response: We thank the reviewer for raising this point, which is a problem worthy of our serious consideration. Unfortunately, we did not conduct the dephosphorylation experiment in this study. We will be more careful for selecting and verifying antibodies.

  1. The issue of describing "activated" microglia compared to resting microglia should be addressed with another marker, or at least with justification.  IBA does not describe the microglia phenotype.

Response: We thank the reviewer for valuable comments. Iba1 is specifically and constitutively expressed in all microglia. The most widely used markers for microglia are Iba1, CD68, CD11b, CD14, CD 45, CD80, CD115, CX3CR1, F4/80, TMEM119, and P2Y12R. Most of them are expressed by both resting and activating microglia. Although Iba1 is specifically and constitutively expressed in all microglia, it is difficult to distinguish the resting and activated microglia, as the reviewer indicated. The recognition of these two phenotypes is possible when comparing amounts of detectable expression levels (Front Cell Neurosci 14:198, 2020). It is also possible to compare resting and activating microglia by electron microscopy (J Neuroinflammation 16:87, 2019). In this study, we applied the transmission electron microscope and found peripheral vacuole-like structures and dark granular bodies in the microglia (representing activated microglia) of 6-month-old AppNL-G-F mice (Figure 4D).

Reviewer 2 Report

Major comments

In this manuscript, the authors investigated the progression of Alzheimer’s disease (AD) with sex differences using a mouse model of AD, by focusing on synaptic dysfunction and neuroinflammation. The authors also examined the therapeutic potentials of Dioscin (Dio) as well as those of Memantine hydrochloride (Mema). The findings are potentially interesting and important; however, there are issues of serious concerns.

  1. Levels of neuroinflammation;

The authors described that they showed progressive elevation of neuroinflammation in young adult AD mouse models (lines 446–448). However, there was no data concerning the levels of proinflammatory cytokines (e.g., TNF-α and IL-1β), although the authors examined the activation levels of inflammation-associated transcription factor NF-κB. It is essential to investigate the levels of proinflammatory cytokines, because those levels are indispensable factors for the study concept. These issues should be addressed.

  1. Neuroprotective effects of Mema;

Although the authors examined the neuroprotective effects of Mema throughout the experiments, they did not describe the results of Mema, except for Figure 1. They also showed no discussion about Mema. I recommend that the authors show and discuss potential neuroprotective effects of Dio in comparison with those of Mema.

  1. Potential factors to increase the weight of brain;

The description “A trend of… (lines 381–383)” is overstated. Immunohistochemical and Western blotting data in this study showed that the amount of amyloid-β was increased in a mouse model of AD compared with control, but these data did not support that the increase of brain weight was attributable to the increase of the amount of amyloid-β only. In humans, one amyloid-β-peptide (1-42) has molecular weight 4514.08, which indicates that 6 × 1023 amyloid-β would correspond to 4514.08 g. Based on this, was there any data in this study to estimate the weight of amyloid-β in the brain from a mouse model of AD and control? If there was not experimental data to quantify the weight of amyloid-β in the brain, I suggest that the authors revise the sentence.

  1. Size of images and graphs;

Size of images and graphs was too small to recognize their contents. I suggest that this should be improved.

Minor comments

  1. I suggest that the citation style on line 94 is revised.
  2. “H2O2” on line 151 should be revised.
  3. Did Figure 1C, E, and G correspond to male? Did Figure 1D, F, and H correspond to female? There was not description about them, and it should be revised.
  4. “Figure 4D” was not in the main text. This should be revised.

Author Response

The response to reviewers’ comments

Reviewer 2

1. Levels of neuroinflammation;

The authors described that they showed progressive elevation of neuroinflammation in young adult AD mouse models (lines 446–448). However, there was no data concerning the levels of proinflammatory cytokines (e.g., TNF-α and IL-1β), although the authors examined the activation levels of inflammation-associated transcription factor NF-κB. It is essential to investigate the levels of proinflammatory cytokines, because those levels are indispensable factors for the study concept. These issues should be addressed.

Response: We thank the reviewer for the important comments and suggestions. We did not measure the expression of the pro-inflammatory cytokines in the brain. However, we examined the serum pro-inflammatory cytokines TNF-α and IL-1β levels using ELISA method. We added this part in the Materials and Results sections, and Figure 5A. We found that the pro-inflammatory cytokine levels increased in AppNL-G-F mice. Administration with Dioscin and Mema inhibited the elevation of the pro-inflammatory cytokine levels.

2. Neuroprotective effects of Mema;

Although the authors examined the neuroprotective effects of Mema throughout the experiments, they did not describe the results of Mema, except for Figure 1. They also showed no discussion about Mema. I recommend that the authors show and discuss potential neuroprotective effects of Dio in comparison with those of Mema.

Response: We thank the reviewer for raising this point. Mema is a non-competitive N-methyl-D-aspartate receptor antagonist used for treating Alzheimer's disease. Animal studies showed that Mema could counteract pathological alterations in a preclinical mouse model of Alzheimer's disease. However, its use is limited by side effects, including dizziness, headache and confusion. In the present study, we examined the effects of Dioscin compared with Mema. The comparison of Dioscin and Mema was added in the Results and Discussion sections, as the reviewer suggested.

3. Potential factors to increase the weight of brain;

The description “A trend of… (lines 381–383)” is overstated. Immunohistochemical and Western blotting data in this study showed that the amount of amyloid-β was increased in a mouse model of AD compared with control, but these data did not support that the increase of brain weight was attributable to the increase of the amount of amyloid-β only. In humans, one amyloid-β-peptide (1-42) has molecular weight 4514.08, which indicates that 6 × 1023 amyloid-β would correspond to 4514.08 g. Based on this, was there any data in this study to estimate the weight of amyloid-β in the brain from a mouse model of AD and control? If there was not experimental data to quantify the weight of amyloid-β in the brain, I suggest that the authors revise the sentence.

Response: We thank you for the helpful comments. As the reviewer indicated, it is difficult to explain the brain weight by the accumulation of amyloid-β in the brain. According to the reviewer’s suggestion, we deleted the sentence (lines 381–383).

4. Size of images and graphs;

Size of images and graphs was too small to recognize their contents. I suggest that this should be improved.

Response: We thank the reviewer for the critical comments. All images were magnified in the manuscript, according to the reviewer’s suggestions.

Minor comments

1. I suggest that the citation style on line 94 is revised.

Thanks. We revised the citation style from ‘Saito et al., 2014’ to ‘ [15] ’.

2. “H2O2” on line 151 should be revised. 

Thanks. ‘H2O2’ was revised to ‘H2O2’.

3. Did Figure 1C, E, and G correspond to male? Did Figure 1D, F, and H correspond to female? There was not description about them, and it should be revised.

Thanks.

4. “Figure 4D” was not in the main text. This should be revised.

Thanks. We added “Figure 4D” in the main text.

Round 2

Reviewer 2 Report

I think that the authors almost appropriately responded to comments raised previously, and the revised manuscript is improved. However, there are issues that need to be addressed;

  1. The description on lines 358–359 (Treatment with…) would be misleading, and it should be revised. The data in the new Figure 5A showed that Dio and Mema significantly suppressed the serum levels of TNF-α, but they did not exhibit significant suppressive effects on those of IL-1β. The authors should describe the data exactly.
  2. Legends for the new Figure 5A should be provided.

Author Response

The response to reviewer’s comments

Reviewer 2

I think that the authors almost appropriately responded to comments raised previously, and the revised manuscript is improved. However, there are issues that need to be addressed.

  1. The description on lines 358–359 (Treatment with…) would be misleading, and it should be revised. The data in the new Figure 5A showed that Dio and Mema significantly suppressed the serum levels of TNF-α, but they did not exhibit significant suppressive effects on those of IL-1β. The authors should describe the data exactly.

Response: We thank the reviewer for the critical comments.

According to the reviewer’s suggestion, we revised the sentence on lines 358-359 as follows:

“Treatment with Dio and Mema inhibited the elevation of the serum TNF-α levels in 6-month-old female mice, but they did not exhibit significant suppressive effects on IL-1b levels”.

  1. Legends for the new Figure 5A should be provided.

Response: We thank the reviewer for the helpful comments.

We added Legends for Figure 5A, as the reviewer indicated.